# The Development of Mitochondrial Gene Editing Tools and Their Possible Roles in Crop Improvement for Future Agriculture

*Jinghua Yang,\* Xiaodong Yang, Tongbing Su, Zhongyuan Hu, and Mingfang Zhang\**

We are living in the era of genome editing. Nowadays, targeted editing of the plant nuclear DNA is prevalent in basic biological research and crop improvement since its first establishment a decade ago. However, achieving the same accomplishment for the plant mitochondrial genome has long been deemed impossible. Recently, the pioneer studies on editing plant mitogenome have been done using the mitochondria-targeted transcription activator-like effector nucleases (mitoTALENs) in rice, rapeseed, and Arabidopsis. It is well documented that mitochondria play essential roles in plant development and stress tolerance, particularly, in cytoplasmic male sterility widely used in production of hybrids. The success of mitochondrial genome editing enables studying the fundamentals of mitochondrial genome. Furthermore, mitochondrial RNA editing (mostly by nuclear-encoded pentatricopeptide repeat (PPR) proteins) in a sequence-specific manner can simultaneously change the production of translatable mitochondrial mRNA. Moreover, direct editing of the nuclear-encoding mitochondria-targeted factors required for plant mitochondrial genome dynamics and recombination may facilitate genetic manipulation of plant mitochondria. Here, the present state of knowledge on editing the plant mitochondrial genome is reviewed.

J. Yang, Z. Hu, M. Zhang
Hainan Institute, Zhejiang University
Yazhou Bay Science and Technology City
Sanya 572025, China
E-mail: yangjinghua@zju.edu.cn; mfzhang@zju.edu.cn

J. Yang, Z. Hu, M. Zhang
Laboratory of Germplasm Innovation and Molecular Breeding
Institute of Vegetable Science
Zhejiang University
Hangzhou 310058, China

X. Yang
Departments of Biology and Plant Science
The Pennsylvania State University
University Park, PA 16802, USA

T. Su
Beijing Vegetable Research Center
Beijing Academy of Agriculture and Forestry Sciences
Beijing 100097, China

 The ORCID identification number(s) for the author(s) of this article can be found under https://doi.org/10.1002/ggn2.202100019

## 1. Introduction

Nowadays, crop production worldwide is facing many challenges, such as rapid population increase, dramatic environmental and climate changes, herbicidal and pesticidal abuse, and pollution from industrial wastes. Agronomists and breeders are putting enormous resources into crop improvement to encounter these challenges. But until recently, such efforts are mainly have been made by the conventional breeding strategies. The application of the CRISPR/Cas9 (clustered regularly interspaced short palindromic repeats/Cas9) system in plants in recent years has paved a new avenue for crop improvement. Although nuclear genome editing accounts for most majority of the gene editing attempts in plants, mitochondrial gene editing tools are also developed and applied in the crop improvement, as such tools have their merits. In this article, we will walk the authors through the history, molecular basis, and application of major mitochondrial gene editing approaches and their possible roles in crop improvement for future agriculture.

Mitochondria are specialized organelles derived from endosymbiosis in eukaryotic cells.[1] Mitochondria are the hub of energy supply chains, biosynthesis center for different types of lipids, and play important roles in the programmed cell death (apoptosis), calcium and redox homeostasis, the reaction to stressors, and nucleic gene expression regulation.[2] Mitochondria genomes underwent many changes during the endosymbiosis process, but a reduced set of protein-coding genes remains and are replicated, transcribed and translated independently from the nuclear genome.[3] In general, mitochondrial genomes are double-stranded DNA molecules but range in size and differ in architecture (circular, linear, or branched) in different species.[4]

Mitochondrial genomes in flowering plants differ in size from about 200 kb to 11 Mb, and their circular, linear, and branched structures are distinct from those in mammalian and yeast cells.[5,6] The long repeat sequences involved in homologous recombination give rise to the dynamics of mitochondrial genome, with a potential to generate subgenomic forms and extensive genomic variation even within the same species.[5] Such changes in genome structure are responsible for the rapid evolution of

plant mitochondrial DNA and for the variants associated with cytoplasmic male sterility and abnormal growth phenotypes.[7–9] In contrast to the variations in mtDNA sizes, the number of mitochondrial genes is relatively conserved in the plant kingdom, with about 60–70 genes found in the mitogenomes of different terrestrial plant species. Most of mitochondrial DNA is non-protein coding.[5] However, most mitochondria encoded genes are critical for the mitochondria functions and the normal development in plants.

To better understand the functions of mitochondrial genes, the mitochondrial and nuclear interaction, and the evolution of mitochondrial genome, the intensive efforts have been put into the development of methodologies for stable plant mitochondria genome editing. Here, we review the exciting efforts and the most recent advancement in the mitochondrial genome editing research. We cover three major areas: precise editing of mitochondrial genomic DNA, mitochondrial RNA editing and editing of the nuclear-encoded genes targeting mitochondria.

## 2. Mitochondrial DNA Editing by Genome Editing Tools

Genomic DNA editing has been widely used in plants for characterizing gene function and crop improvement, especially in recent years with the application of the CRISPR/Cas9 system in plants.[10] While it is fairly easy to introduce the targeted modification to the nuclear genome by using CRISPR/Cas9, targeted mitochondrial genome editing in plants remains very challenging. The application of the CRISPR/Cas9 system in mitochondrial genome editing is complicated by the double membrane of mitochondria, and it is very difficult for the guide RNA to cross the inner mitochondrial membrane due to the strong electrochemical potential present across the membrane.[11] Stable transformation of mitochondrial genomes using the mitoCRISPR-Cas approach was not achieved until recently. Hussain et al. were able to perform gene editing in the mitochondrial DNA by using the CRISPR/Cas9 system; they reduced the expression of the mitochondrial-encoded gene ND4 (NADH-ubiquinone oxidoreductase chain 4) by appending an ND4-targeting guide RNA to an RNA transport-derived stem loop element (RP-loop) and expressing the Cas9 enzyme with a preceding mitochondrial localization sequence in the murine cell cultures.[12]

The first case of directed double-strand breaks in plant mitochondrial genomes using mitochondria-targeted TALENs in cytoplasmic male sterile rice and rapeseed was achieved by Arimura and his colleagues.[13] Subsequently, stable genomic editing of the mitochondrial genome using mitoTALENs was established in model plant Arabidopsis.[14] Applying mtDNA editing in Arabidopsis is difficult because most mitochondria-encoded genes in Arabidopsis are the single-copy and essential genes, and knocking out any of them would cause embryo lethality. To circumvent this issue, Arimura et al chose to work on ATP synthase 6 (a small subunit of F1Fo ATP synthase), a protein encoded by two gene copies, *ATP6-1* and *ATP6-2*, in *Arabidopsis thaliana* Columbia-0 (Col-0). They successfully knocked out either *ATP6-1* or *ATP6-2* by using mitoTALEN vectors via floral dipping. They also pointed out the importance of avoiding the influence of nuclear copies of mitochondrial DNA (Numts) when designing the confirmation primers after the transformation.[15,16]

It is exciting to see the mitoTALEN approach eventually used in the model plants, but, this system has its own limits. The linearized mtDNA generated by fusion degrades rapidly; consequently, the modified mtDNA copies will eventually be outnumbered by the uncut version. The other issue is the mitoTALENs or mitoZFNs approaches cannot introduce specific nucleotide changes.[17] However, the precise gene editing of mitochondrial DNA (mtDNA) has been reported recently.[18] This new method utilizes a novel bacterial cytidine deaminase, named "dsDNA deaminase toxin A", or DddA, which specifically convert cytosines to thymines (C→T) present in double-strand DNA. This is achieved by adding DddA to the MTS-TALEN construct, forming MTS-TALEN-DddA fusion proteins, named "DddA-derived cytosine base editor" (DdCBE). The Mok group tested different combinations of MTS-TALEN-DddA fusion vectors and precisely edited at least six different mitochondrial genes in the human cell lines, with efficiencies ranging from 4.6% to 49%.[11] This breakthrough is going to accelerate the mitochondrial genome research in the animal model systems. The usage of MTS-TALEN-DddA has been tested in the plants as well. Recently, Kang et al. have assembled 16 DdCBE plasmids and used the resulting DdCBEs to efficiently introduced point mutagenesis in the mitochondria and chloroplasts of lettuce or rapeseed calli,[19] we can expect this system will serve as a valuable resource for future organelle DNA function research and potential application of mitochondrial DNA editing in the crop improvement. As mitochondria play a critical role in the plant biotic and abiotic stress response, redox homeostasis, and phytohormone regulation.[20] Targeted mitochondrial gene editing could create specific retrograding signaling and trigger downstream gene expression changes, and eventually create novel phenotypes.

## 3. Mitochondrial RNA Editing by Designed Pentatricopeptide Repeat Proteins

In addition to mitochondrial DNA editing, it is also possible to introduce or reverse mitochondria-associated traits by mitochondrial RNA editing, which can be achieved by RNA-binding pentatricopeptide repeat (PPR) proteins to target editing sites in a sequence-specific manner. RNA editing is based on intrinsic biological machinery that exists in many eukaryotic species.[21,22] It is a process that changes the bases of RNA at the posttranscriptional level. So far, at least 400 editing sites have been discovered in Arabidopsis mitochondrial RNAs,[23] with most of them located in the protein-coding gene regions and only a few sites in the non-coding regions.[24] In plants, natural RNA editing is carried out by a protein complex called editosome, a complex that contains multiple protein factors with the PPR proteins as its core components. The PPR proteins carry a tandem array of 35-aminoacid helix-turn-helix motifs[25] that can be classified into two subgroups: PPR-P and PPR-PLS.[22,26,27] These two subgroups are distinguished by their structures and the functions they carry out. The PPR-P subclass plays a crucial role in a wide range of organellar RNA metabolism, whereas the PPR-PLS subclass mainly involves C-U RNA editing.[28] Many of the PLS proteins contain three types of highly conserved protein domain modules (E1, E2, and DYW) at their carboxy terminus.[26] The DYW domain named after the conserved amino acids aspartate (D), tyrosine (Y), and tryptophan (W), contains sequences similar

to a conserved cytidine/deoxycytidylate deaminase motif.[29] The sequence specificity of RNA editing is achieved by the PPR array recognizing the target RNA, and the DYW domain catalyzing the C-to-U conversion.

Based the molecular basis of natural mitochondrial RNA editing, So far, at least three approaches have been developed to achieve precise designed mitochondrial RNA editing: 1) RNA editing factors (PLS-type with DYW domains) engineering. The sequence specificity and the one-to-one relationship between the DYW-type PPRs and their targets enables the designed precise C-to-U modification in the targeted plant mitochondrial transcripts.[26,30–32] It was demonstrated for the first time in an individual moss (*Physcomitrella patens*) that a DYW-type PPR protein alone can introduce efficient C-to-U editing in *Escherichia coli* and reproduce the moss mitochondrial editing.[33] Using this straightforward *E. coli* system, different types of designed DYW-type PPRs from different species were tested. They expressed the *P. patens* editing factor PPR79 and successfully edited all target sites in the Arabidopsis cwm1 mutant background; the homolog of PPR79 in Arabidopsis is CWM1.[34] 2) P-type PPR engineering. Precise mitochondrial RNA editing might also be achieved by P-type PPR (non-editing factors) protein engineering. The first example of a targeted block in the expression of a specific mitochondrial transcript by a custom-designed PPR protein was reported by Colas des Francs-Small et al, whereby they used a modified PPR protein RNA PROCESSING FACTOR 2 (RPF2) to eliminate the expression of the mitochondrial NAD6 gene in *A. thaliana*.[35] 3), RNA editing using CRISPR-Cas system. Although designed RNA editing can also be achieved by using the CRISPR-Cas system,[36,37] so far in plants, no successful targeted mitochondrial RNA editing via CRISPR-Cas has been reported. Just as in DNA editing via the mitoCRISPR-Cas, the challenge that CRISPR-Cas-mediated mitochondrial RNA editing faces is to introduce an efficient mean to deliver the gRNA to the mitochondrial matrix. The development of bioinformatics also fueled the research on the targeted mitochondrial RNA editing. A complete list of bioinformatics tools and databases that can be used for RNA editing study and engineering is included in the Giudice's comprehensive review.[38] Considering all of these developments in methodologies for targeted mitochondrial RNA editing, and a wide range of important plant traits for RNA editing, we believe the targeted mitochondrial RNA editing approach opens a novel avenue for reverse genetics studies of mitochondrial gene function and will eventually have great applications in crop breeding.

## 4. Editing the Nuclear-Encoded Genes Required for Mitochondrial Surveillance

As mentioned, the plant mtDNAs have quite complicated and dynamic structures and can form a set of subgenomic forms that are permutable by recombination. Therefore, surveillance machinery is developed to control mtDNA mutation rates and to maintain genome stability. In plants, this surveillance is carried out by a group of proteins involved in DNA replication and mismatch repair. Some examples are *MSH1*, *RecA2*, *RecA3*, *WHY2*, *OSB1*, and *RECG1*.

The MSH1, MUTS HOMOLOG 1 is dual targeted to both chloroplasts and mitochondria. It was the first protein described as an mtDNA mutator in Arabidopsis.[39] It functions in mismatch repair as an inhibitor of strand invasion during homologous recombination.[40] The MSH1 has been shown to limit ectopic recombination in plant cytoplasmic genomes and maintain the stability of the mitochondria genome in higher plants.[41] It was reported that disrupting the MSH1 (but not the other mtDNA repair genes) leads to massive increases in the frequency of point mutations and small indels (approximately 10- to 1000-fold) and changes to the mutation spectrum in mitochondrial DNA.[42] Disruption in the *MSH1* gene resulted in enhanced mitochondrial genome recombination at numerous repeated sequences, and a range of distinct phenotypes in five species including Arabidopsis, sorghum, tobacco, tomato, and pearl millet.[43] Further investigation revealed that the mitochondrial genome rearrangements associated with the MSH1 depletion in mitochondria resulted in enhanced heat tolerance and the curly and wrinkled leaf phenotypes observed in the *msh1* mutant. Hemi-complementation of *msh1* mutants with MSH1 specifically targeted to mitochondria completely restored mtDNA stability and the curly or wrinkled leaf phenotypes.[44] In *Brassica juncea*, the MSH1 mediated higher substoichiometric shifting (SSS) activity and increased in the *ORF220* copy number to induce male sterility and fertility reversion.[45,46] Plant mitochondria possess two eubacteria-type RecA proteins that are core components of the mitochondrial repair mechanisms. In Arabidopsis, the *recA2* and *recA3* mutations trigger increased ectopic mitochondrial DNA recombination. The *recA2* is lethal at the early seedling stage, whereas the *recA3* first-generation mutant plants are normal in most cases, but develop more severe phenotypes and became hypersensitive to genotoxic treatments in the later mutant generations.[47,48] These results suggest it is possible to introduce changes to the mitochondrial genome via editing the nuclear-encoded mitochondria-targeted genes and eventually create new traits in plants.

## 5. Perspective

Although much is known about the genomic sequences of plant mitochondria, the attempts to directly edit the mitochondrial genome using CRISPR/Cas systems have been hindered by the challenge of transporting the gRNA into the mitochondrial matrix.[49] Nevertheless, the development of CRISPR tools for mitochondrial genome editing has been described in mammalian cells, with controversial results because of the efficiency of import into mitochondria.[50,51] It was demonstrated that gRNA could be imported efficiently into the mitochondria of mammalian cells with an additional 20-bp stem-loop element of nuclear RNase P.[12,52] More recently, an RNA-free base editor has been developed for human cells by fusing a non-toxic half of an interbacterial toxin with the transcription activator-like effector array proteins and an uracil glycosylase inhibitor to edit precisely the mitochondrial genome.[18] This raises the possibility of precisely manipulating mitochondrial genomes in plants. As the MTS-TALEN-DddA DNA editing already been tested in plants,[19] we would expect decent amount of research using this technology to study mitochondrial gene function. In addition, it would be exciting to see the mitoCRISPR-Cas DNA editing get applied in plants in the future.[53] The development of more mitochondrial DNAs editing approaches will help us reach a better understanding of

**2100019 (3 of 6)**

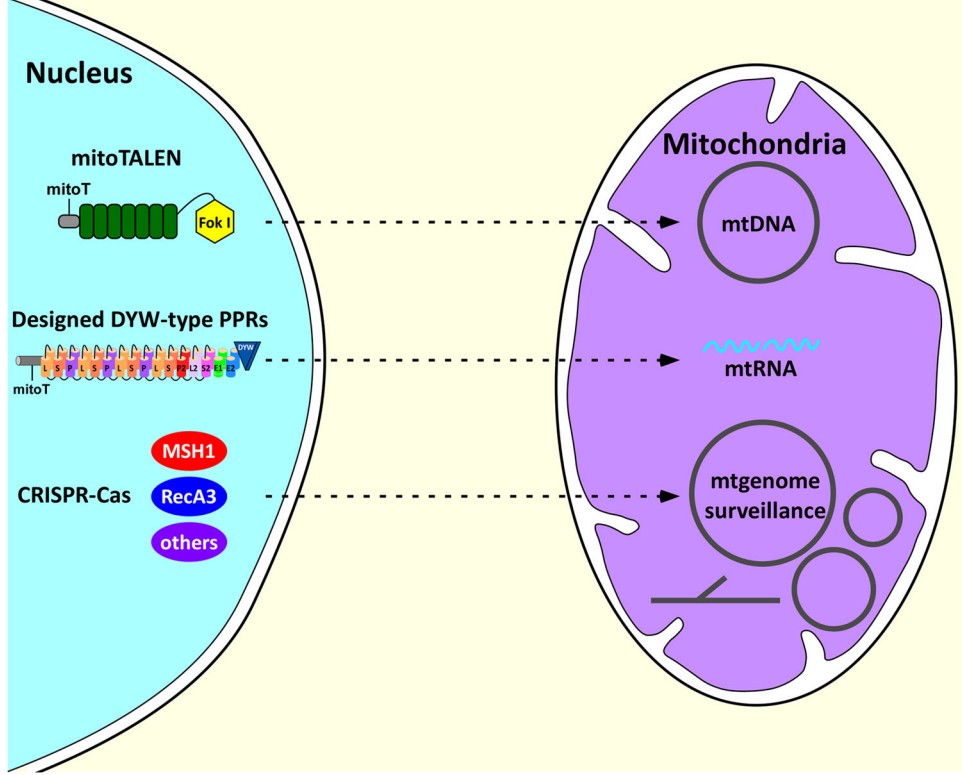

**Figure 1.** Approaches to achieve the plant mitochondrial genome editing. MitoTALEN is used to precisely edit CMS-associated ORFs in the CMS plant genome and mitochondrial genes in Arabidopsis genome. The schematic model of mitoTALEN is detailed in the publication.[58] Designed PPRs are able to directly target mitochondrial RNAs to regulate mitochondrial gene expression. The schematic model of PPRs is described the publication.[22] Editing of nuclear-encoded and mitochondria-targeted genes, including *MSH1*, *RecA3*, and others, can alter the mitochondrial genome surveillance.

the cross-talk between the nuclear and mitochondrial genome, and to eventually get applied to the creation of novel germplasms for crop improvement. More specifically, we expect mitochondrial DNA editing approaches will play an important role in creating abiotic and biotic stress tolerant plant germplasms in the future. We can also expect that the designed mitochondrial-targeted PPR proteins will be used more frequently and efficiently in mitochondrial gene function studies. The feasibility of designed PPR proteins to bind specifically to a wide array of mitochondrial RNA sequences will be enhanced. The range of species this technology can be applied to will expand.

The development of mitochondrial genome editing methodologies will benefit the plant breeding efforts; actually, we have already started to see such examples. Kazama et al managed to cure cytoplasmic male sterility in rice and rapeseed by using mitoTALEN to knock out the CMS-associated genes in the CMS line.[13,54] Moreover, the fertility reversion rate of the CMS line in *Brassica juncea* can be altered by manipulating the MSH1 expression;[46] In both cases, the mitochondrial genome-modified materials can facilitate the utilization of heterosis, several studies reported that *msh1* mutation derived epigenetic variation can be selected for crop improvement, produce plant materials with growth vigor and abiotic tolerance in tomato,[55,56] soybean,[57] and sorghum.[55] Although to what extent mitochondria retrograde signaling contribute to these epigenetic variations still remain unclear, these findings suggest that editing the nuclear-encoded mitochondria-targeting genes could have agricultural applications. We believe the ongoing improvement in the mitochondrial genome editing methodologies will eventually lead to more advances in crop improvement, as the mitochondria genome manipulation used to introduce novel traits, or increase the plant resilience to emerging agricultural challenges. One thing is for sure, this is going to be an exciting era for scientists studying mitochondrial biology in plants (**Figure 1**).

## Supporting Information

Supporting Information is available from the Wiley Online Library or from the author.

## Acknowledgements

This work was supported in part by grants from the Hainan Provincial Joint Project of Sanya Yazhou Bay Science and Technology City (320LH003), the Key Research and Development Project of Hainan Province, the National Natural Science Foundation of China (32030092, 32172558, 31872095, 31372063, 32172557) and the National Natural Science Foundation of Zhejiang Province (LZ20C150002).

## Conflict of Interest

The authors declare no conflicts interest.

# ADVANCED SCIENCE NEWS

www.advancedsciencenews.com

# ADVANCED GENETICS

www.advgenet.com

## Author Contributions

J.Y.: Conceptualization; writing-original draft; writing-review and editing. X.Y.: Conceptualization; writing-original draft; writing-review and editing. T.S.: Writing-review and editing. Z.H.: Writing-review and editing. M.Z.: Supervision; writing-review and editing.

## Peer Review

The peer review history for this article is available in the Supporting Information for this article.

## Keywords

genome editing, mitoTALEN, nuclear-encoding mitochondria-targeted factors, pentatricopeptide repeat proteins, plant mitochondrial genome

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

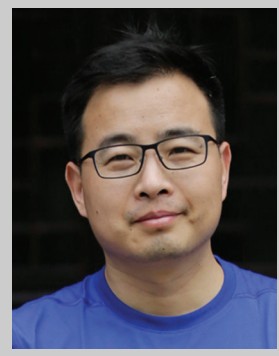

**Jinghua Yang** is a professor of College of Agriculture and Biotechnology, Zhejiang University, China. Dr. Yang obtained his Ph.D. degree in vegetable science from Zhejiang University in 2006. He did his post-doctoral research at the Laboratory of Plant Molecular and Genetics, the University of Tokyo, Japan. His lab works on germplasm innovation and breeding for vegetable crops using genomics, genetics and genome editing approaches, mainly interested in genome evolution and trait selection, vegetative organ development and quality formation, RNA viruses, and host interaction and cytoplasmic male sterility.

