## [**Supplementary Information**: Record of Transparent Peer Review · Advanced Genetics]

Record of Transparent Peer Review

The development of mitochondrial gene editing tools and their possible roles in crop improvement for future agriculture

Jinghua Yang*, Yiaodong Yang, Tongbing Su, Zhongyuan Hu, Mingfang Zhang*

*Corresponding

Review timeline:

Date Submitted: 01-Jul-2021
Editorial Decision: 04-Aug-2021 Minor Revision
Revision Received: 18-Aug-2021
Editorial Decision: 18-Aug-2021 Accept in Principle
Accepted: 20-Aug-2021

Editor: Myles Axton

Initial Editorial Evaluation	01-Jul-2021
-------------

Summary

While genome editing of the nucleus is established for a decade and now used in plant breeding, organelle editing is more difficult. Given essential roles in crop yield and basic plant biology, as well as technical use in creating male sterile lines for hybrid breeding, mitochondrial editing is now feasible and useful in understanding crop biology and optimizing breeding. Topics include "precise editing of mitochondrial genomic DNA, mitochondrial RNA editing and editing of the nuclear-encoded genes targeting mitochondria."

Scope

Do the research, methods or topics fit within the aims of this, or another journal?

Yes, the use of DNA and RNA editing in plant breeding and analysis of plant development is of great interest to Advanced Genetics.

1 st Peer Review	02-Jul-2021 to 04-Aug-2021
----------------------------

Reviewer #1

The manuscript entitled "Editing the plant mitochondrial genome enables variations for crop improvement" by Jinghua Yang and his coauthors is review on the study of (plant) mitochondria genome editing. As note in the title, the authors aim to summarize the progress made in mitochondria genome editing and its importance to future crop improvement. The summary is suitable for publication and I have mostly minor comments for the authors.

1.1 I suggest to change the title to something catchier, like "The development of mitochondrial gene editing tools and their possible roles in crop improvement for future agriculture".

In that aspect and based on their title, the authors need to emphasis in the first paragraph of the Intro part the importance of genome editing to plant improvement in light of climate changes, herbicidal and pesticidal uses (as e.g. BT toxin, glyphosates), as so far, the vast majority of crops have been made by "non-GMO" breeding strategies. What is the added value and importance of mtDNA add to that of the on-going efforts of nuclear gene transfer? (maybe one possibility is higher expression or the fact that these organelles are very often transmitted only maternally?!)

I found several language mistakes and syntax errors that need to be fixed before publication. Some examples are found here:

1m1. Intro first paragraph: as outlined above start with the importance of genome editing to plant improvement

1m2. Page 4 line 17: "Scientists often say we are living in the era of genome editing." Consider to rephrase

1m3. Page 4 Line 20: replace feat with fate?

1m4. Page 4 Line 21: Mitochondrial DNA = mitogenome, or mtDNA

1m5. Page 4 Line 27: remove the in the quotes

1m6. Page 5 Lines 46-48: nuclear genome (3). In general, mitochondrial genomes are double-stranded DNA molecules differing in size and architecture (circular, linear or branched forms in different species) (4) - consider to rephrase

1m7. Page 5 Line 49: instead of “differ” in size write range in size.

1m8. Page 5 Line 57: instead of “varied genome size”, consider “variations in mtDNA sizes”

1m9. Page 5 Line 58: instead of “known genes found in different terrestrial” consider “ found in the mitogenomes of different...”

1m10. Page 6 Lines 63-64: “mitochondrial genome, the enormous efforts” consider “intensive” instead..

1m11. Page 6 Line 104: replace it with “this system”

1m12. Page 10 line 160: “how to find a reliable method to deliver gRNA to the mitochondrial..” consider “to introduce an efficient mean to deliver the gRNA..”

1m13. Also the authors need to refer to the MSs: Sultan et al (2016) Plant cell, 28, 2805-2829 and Val et al (2011) were the authors used tRNA-like ribozymes to be imported into plant mitochondria!

1m14. Page 12 line 192: mutant names (i.e. *msh1*) should be in Italic

Reviewer #2

Review for Advanced Genetics paper (Review GGN-2021-0011):

The paper ‘Editing the plant mitochondrial genome enables variations for crop improvement’ by Yang et al. reviews most attempts of editing the plant mitochondrial genome, from mitochondrial DNA editing to targeted mitochondrial RNA editing, and that of nuclear factors involved in the mitochondrial genome dynamics and recombination.

2.1 Most relevant papers published so far are cited, except a newly published paper on the use of the ‘DddA-derived cytosine base editor’ (DdCBE) in plants, which should be added at the end of the paragraph treating Mitochondrial DNA editing (line 117).

Kang, BC., Bae, SJ., Lee, S. et al. Chloroplast and mitochondrial DNA editing in plants. Nat. Plants 7, 899–905 (2021). <https://doi.org/10.1038/s41477-021-00943-9>

2.2 My biggest criticism is that, although this review is concise, it is not very helpful as the text lacks clarity and structure. This may be due to an imprecise command of the English language, but the whole text is rather confusing.

- Has the author represented the main concepts and advances in the field fairly?

I think the description of the concepts and advances lacks precision and the information can be confusing in places.

Section 1: the introduction is generally fine.

Section 2 - the section about mtDNA editing is a bit confusing because the examples are alternatively from mammalian or plant cells, and there are substantial differences between them. I would prefer seeing a table summarising/comparing the different approaches tried in mammalian cells and in plants, and stressing the problems encountered.

Section 3 is very confusing because the authors mix different phenomena under the same name: in this section about “RNA editing”, the authors describe the natural C → U or U → C RNA editing process performed by nuclear factors at a post-transcriptional level on mitochondrial and plastid RNAs by editing factors (PLS-type with DYW domains) and then cite the example of the “targeted modification of mitochondrial transcripts”, which they also name RNA editing, using the P-type PPR protein RPF2 (which is not an editing factor). They then talk about the RNA editing assay performed by Physcomitrella editing factors by Oldenkott and collaborators in *E. coli*. This section should be rewritten, stressing what the purpose of the

experiments was in each case (targeted modification of mitochondrial transcript expression, or straight editing assay etc...) and discuss their possible applications.

Section 4: The authors do not explain how manipulating nuclear factors involved in the mitochondrial genome dynamics and recombination could help precise editing the plant mitochondrial genome for crop improvement.

2.3 Section 5 (numbered 4 in the MS) is called perspectives but is more a summary of previous sections and has very few realistic perspectives as all these techniques are not really applicable to crop improvement at this stage. It should be improved.

- How will this Perspective lead researchers to conduct their research differently?

I don't think this review is very helpful as it is.

- Are the claims, evidence and views presented in a clear and logical order?

Although the sections are presented in a logical order in the manuscript (DNA editing, "targeted mitochondrial transcript editing", nuclear genes), I think this review is very confusing because the contents of each sections are not sufficiently structured.

Minor points:

2m1- Line 79-85, please specify that this was done in murine cell cultures: "Hussain et al were able to perform gene editing of the mouse mitochondrial DNA in cell cultures"

2m2- Line 93: "F1F0 ATP synthase" should be "F1Fo ATP synthase")

2m3- Line 99: Nuclear mitochondrial DNA (NUMT) result from duplication of some regions of the mitochondrial DNA which are integrated to the nuclear genome, but are not, at least in the case of Arabidopsis thaliana, expressed. It is therefore very imprecise to describe this phenomenon as "During co-evolution of nuclear and mitochondrial genomes, the nuclear genome sometimes may develop pseudo-genes that have a high similarity to true mitochondrial genes". I think this should be rephrased. Nevertheless, this comment (line 96-102) is not very relevant in the context of this review.

2m4- Please check the formatting of the bibliography (here are just two examples):

16. Arimura S. Effects of mitoTALENs-directed double-strand breaks on plant mitochondrial genomes comment. Genes-Basel. 2021;12(2).

31. des Francs-Small CC, Sanglard LVP, Small I. Targeted cleavage of nad6 mRNA induced by a modified pentatricopeptide repeat protein in plant mitochondria. Commun Biol. 2018;1.

Should be:

31. Colas des Francs-Small C, Sanglard LVP, Small I. Targeted cleavage of nad6 mRNA induced by a modified pentatricopeptide repeat protein in plant mitochondria. Commun Biol. 2018;1.

1 st Editorial Decision	04-Aug-2021
-------------

Editorial decision: Minor Revision incorporating all the reviewers' suggestions

Editor's understanding of the reviews

Reviewer #1 Recommends Minor Revision

Reviewer #2 Recommends Major Revision

Author's Response to 1 st Review	18-Aug-2021
---	-------------

Reviewer comments	Editor recommendation	Author reply	Changes to Manuscript
2.2 although this review is concise, it is not very helpful as the text lacks clarity and structure.... the description of the concepts and	ED1 Organize the overall Review Article carefully and within particularly section 3 distinguish the different kinds of RNA editing and	We rewrite this section to address reviewer's concern	The Section 3 is rewritten as reviewer and editor suggested. We distinguished natural mitochondrial RNA editing

advances lacks precision and the information can be confusing.... Section 3 is very confusing.... the contents of each of the sections are not sufficiently structured.	explain which of these can be used for crop improvement.		from designed mitochondrial RNA editing, and categorized designed mitochondrial RNA editing into three approaches: 1) RNA editing factors (PLS-type with DYW domains) engineering. 2) P-type PPR engineering. 3), RNA editing using CRISPR-Cas system.
2.3 Section 5 (numbered 4 in the MS) is called perspectives but is more a summary of previous sections and has very few realistic perspectives as all these techniques are not really applicable to crop improvement at this stage.	ED2 Explain specifically how each of the techniques has potential to contribute to crop improvement. What can not be done by mitochondrial editing, and can to what extent might nuclear editing or RNA manipulation replace this technology?	Both Kazama et al and Zhao et al cited are examples that the mitochondrial genome-modified materials can facilitate the utilization of heterosis, and we added more examples and discussion in this section as requested	Added the MTS-TALEN-DddA DNA editing in plant content. Clarified current examples Added the Mackenzie group research on using msh1 for crop improvement Added more discussion
1.1 I suggest to change the title to something catchier, like “The development of mitochondrial gene editing tools and their possible roles in crop improvement for future agriculture”. In that aspect and based on their title, the authors need to emphasis in the first paragraph of the Introduction part the importance of genome editing to plant improvement in light of climate changes, herbicidal and pesticidal uses (as e.g. BT toxin, glyphosates), as so far, the vast majority of crops have been made by “non-GMO” breeding strategies. What is the added value and importance of mtDNA add to that of the on-going efforts of nuclear gene transfer? (maybe one possibility is higher expression or the fact that these organelles are very often transmitted only maternally?!)	ED3 This is an excellent suggested title. The reviewer is right that the Introduction should set up what is possible with nuclear gene editing of crops and explain the need for mitochondrial editing (is plastid editing necessary or feasible?). This Introduction should set up the questions that the Perspectives section (ED2 above) answers and explains how crops can be improved using these new targeting techniques.	Agree	Title changed as suggested Added content to the introduction as suggested

Reviewer #1

The manuscript entitled “Editing the plant mitochondrial genome enables variations for crop improvement” by Jinghua Yang and his coauthors is review on the study of (plant) mitochondria genome editing. As note in the title, the authors aim to

summarize the progress made in mitochondria genome editing and its importance to future crop improvement. The summary is suitable for publication and I have mostly minor comments for the authors.

1.1 I suggest to change the title to something catchier, like “The development of mitochondrial gene editing tools and their possible roles in crop improvement for future agriculture”.

In that aspect and based on their title, the authors need to emphasize in the first paragraph of the Intro part the importance of genome editing to plant improvement in light of climate changes, herbicidal and pesticidal uses (as e.g. BT toxin, glyphosates), as so far, the vast majority of crops have been made by “non-GMO” breeding strategies. What is the added value and importance of mtDNA add to that of the on-going efforts of nuclear gene transfer? (maybe one possibility is higher expression or the fact that these organelles are very often transmitted only maternally?!)

Author Response: Thanks for your valuable comments. We changed the title as suggested in this revised version. And we added new description on gene editing facilitating crop improvement and marked them in red in the text.

I found several language mistakes and syntax errors that need to be fixed before publication. Some examples are found here:

1m1. Intro first paragraph: as outlined above start with the importance of genome editing to plant improvement

1m2. Page 4 line 17: “Scientists often say we are living in the era of genome editing.” Consider to rephrase

1m3. Page 4 Line 20: replace feat with fate?

1m4. Page 4 Line 21: Mitochondrial DNA = mitogenome, or mtDNA

1m5. Page 4 Line 27: remove the in the quotes

1m6. Page 5 Lines 46-48: nuclear genome (3). In general, mitochondrial genomes are double-stranded DNA molecules differing in size and architecture (circular, linear or branched forms in different species) (4) - consider to rephrase

1m7. Page 5 Line 49: instead of “differ” in size write range in size.

1m8. Page 5 Line 57: instead of “varied genome size”, consider “variations in mtDNA sizes”

1m9. Page 5 Line 58: instead of “known genes found in different terrestrial” consider “found in the mitogenomes of different...”

1m10. Page 6 Lines 63-64: “mitochondrial genome, the enormous efforts” consider “intensive” instead..

1m11. Page 6 Line 104: replace it with “this system”

1m12. Page 10 line 160: “how to find a reliable method to deliver gRNA to the mitochondrial..” consider “to introduce an efficient mean to deliver the gRNA..”

1m13. Also the authors need to refer to the MSs: Sultan et al (2016) Plant cell, 28, 2805-2829 and Val et al (2011) were the authors used tRNA-like ribozymes to be imported into plant mitochondria!

1m14. Page 12 line 192: mutant names (i.e. msh1) should be in Italic

Author Response: Revised them as suggested and marked them in red in the text.

Reviewer #2

The paper ‘Editing the plant mitochondrial genome enables variations for crop improvement’ by Yang et al. reviews most attempts of editing the plant mitochondrial genome, from mitochondrial DNA editing to targeted mitochondrial RNA editing, and that of nuclear factors involved in the mitochondrial genome dynamics and recombination.

2.1 Most relevant papers published so far are cited, except a newly published paper on the use of the 'DddA-derived cytosine base editor' (DdCBE) in plants, which should be added at the end of the paragraph treating Mitochondrial DNA editing (line 117).

Kang, BC., Bae, SJ., Lee, S. et al. Chloroplast and mitochondrial DNA editing in plants. Nat. Plants 7, 899–905 (2021).
<https://doi.org/10.1038/s41477-021-00943-9>

Author Response: Thanks for the comments. We did not cite this reference because it is not published when we submitted our MS. In this revised version, we added this citation.

2.2 My biggest criticism is that, although this review is concise, it is not very helpful as the text lacks clarity and structure. This may be due to an imprecise command of the English language, but the whole text is rather confusing.

Author Response: Thanks for the comments. We try our best to improve this MS as possible as we can and asked English editing service from native speaker.

- Has the author represented the main concepts and advances in the field fairly?

I think the description of the concepts and advances lacks precision and the information can be confusing in places.

Author Response: Thanks for the valuable comments. We added some new description in this revised version and marked them in red in the text.

Section 1: the introduction is generally fine.

Section 2 - the section about mtDNA editing is a bit confusing because the examples are alternatively from mammalian or plant cells, and there are substantial differences between them. I would prefer seeing a table summarising/comparing the different approaches tried in mammalian cells and in plants, and stressing the problems encountered.

Section 3 is very confusing because the authors mix different phenomena under the same name: in this section about "RNA editing", the authors describe the natural C → U or U → C RNA editing process performed by nuclear factors at a post-transcriptional level on mitochondrial and plastid RNAs by editing factors (PLS-type with DYW domains) and then cite the example of the "targeted modification of mitochondrial transcripts", which they also name RNA editing, using the P-type PPR protein RPF2 (which is not an editing factor). They then talk about the RNA editing assay performed by *Physcomitrella* editing factors by Oldenkott and collaborators in *E. coli*. This section should be rewritten, stressing what the purpose of the experiments was in each case (targeted modification of mitochondrial transcript expression, or straight editing assay etc...) and discuss their possible applications.

Section 4: The authors do not explain how manipulating nuclear factors involved in the mitochondrial genome dynamics and recombination could help precise editing the plant mitochondrial genome for crop improvement.

2.3 Section 5 (numbered 4 in the MS) is called perspectives but is more a summary of previous sections and has very few realistic perspectives as all these techniques are not really applicable to crop improvement at this stage. It should be improved.

- How will this Perspective lead researchers to conduct their research differently?

I don't think this review is very helpful as it is.

Author Response: We talked about recent approaches towards mitochondrial gene editing for crop improvement in future.

- Are the claims, evidence and views presented in a clear and logical order?

Although the sections are presented in a logical order in the manuscript (DNA editing, "targeted mitochondrial transcript editing", nuclear genes), I think this review is very confusing because the contents of each sections are not sufficiently structured.

Author Response: Thanks for the comments. The three approaches of mitochondrial gene editing including direct mtDNA editing, targeted mitochondrial transcript editing and nuclear genes associated with mitochondrial surveillance are really

belonged to different mechanism and make them to be independent. To date, these three approaches to edit mitochondria possibly worked and could be used for crop improvement in future.

Minor points:

2m1- Line 79-85, please specify that this was done in murine cell cultures: “Hussain et al were able to perform gene editing of the mouse mitochondrial DNA in cell cultures”

Author Response: Rephrased.

2m2- Line 93: “F1F0 ATP synthase” should be “F1Fo ATP synthase”)

Author Response: Corrected as suggested.

2m3- Line 99: Nuclear mitochondrial DNA (NUMT) result from duplication of some regions of the mitochondrial DNA which are integrated to the nuclear genome, but are not, at least in the case of Arabidopsis thaliana, expressed. It is therefore very imprecise to describe this phenomenon as “During co-evolution of nuclear and mitochondrial genomes, the nuclear genome sometimes may develop pseudo-genes that have a high similarity to true mitochondrial genes”. I think this should be rephrased. Nevertheless, this comment (line 96-102) is not very relevant in the context of this review.

Author Response: Related content is deleted as reviewer suggested.

2m4- Please check the formatting of the bibliography (here are just two examples):

16. Arimura S. Effects of mitoTALENs-directed double-strand breaks on plant mitochondrial genomes comment. Genes-Basel. 2021;12(2).

31. des Francs-Small CC, Sanglard LVP, Small I. Targeted cleavage of nad6 mRNA induced by a modified pentatricopeptide repeat protein in plant mitochondria. Commun Biol. 2018;1.

Should be:

31. Colas des Francs-Small C, Sanglard LVP, Small I. Targeted cleavage of nad6 mRNA induced by a modified pentatricopeptide repeat protein in plant mitochondria. Commun Biol. 2018;1.

Author Response: Checked according to the journal style requests.

2 nd Editorial Decision	18-Aug-2021
The manuscript has now been revised incorporating the comments of the reviewers and editor, and we have now decided to accept the revised manuscript in principle, subject to the attached formatting requirements.	

Final Editorial Decision	20-Aug-2021
The article is now formally accepted